# OpenReview forum: "Multi-Marginal Stochastic Flow Matching for Alignment of High-Dimensional Snapshot Data at Irregular Time Points"
_ICLR.cc/2025/Conference — ICLR 2025 Conference Withdrawn Submission_

### Official Review · Reviewer_3yuc · 2024-11-02

**Soundness:** 3
**Presentation:** 2
**Contribution:** 2
**Rating:** 3
**Confidence:** 4

**Summary:**

This paper introduces a method for computing the evolution of high-dimensional snapshot data taken at irregular time points using multi-marginal stochastic flow matching. Aimed at single-cell perturbation studies, it addresses the limitations of traditional approaches that often rely on dimensionality reduction. Instead, this model operates in a high-dimensional space where it interpolates smooth transitions between non-equidistant time points using optimal transport plans. Experimentally, the paper demonstrates the method on a limited set of synthetic and a single real-world single-cell datasets.

**Strengths:**

- **Introduction and Motivation**: The paper is well-motivated and effectively introduces the research problem. Both the abstract and Section 1 are clearly written.
- **Problem Statement**: The problem is clearly stated and directly addresses an existing challenge in practice, particularly within life sciences. It is well-grounded in biological applications and provides relevant context.
- **Approach**: The approach seems to be solid. It directly models flows in high-dimensional space, aims to introduce a certain robustness, and uses overlapping triplets in time-series alignment captures local dynamics at non-equidistant intervals.

**Weaknesses:**

- **Problem Formulation and Methodology**: The explanation in Section 2 could benefit from additional intuitive descriptions, particularly clarifying relationships between indices $i$, $t_1$, and $t_i$. The transition from Section 2.0 to 2.1 lacks motivation, and it is not immediately clear why the authors chose to consider Optimal Transport (OT) in their methodology. The rationale behind some methodological decisions---such as the use of dynamic formulation for Wasserstein distance---needs significant expansion. Defining $u_t(x)$ earlier in the text would also improve clarity.
- **Hermitian Cubic Splines**: The introduction of Hermitian cubic splines seems superfluous for the remainder of the paper, adding complexity without relevancy. Additionally, it is unclear why the transition from basic transport splines to complex OT plan compositions (l. 157) is necessary or beneficial, as it lacks arguments. This part needs a better contextualized to clarify why compositions of OT plans are advantageous.
- **Drift Networks and Score Networks**: In Section 2.3, the introduction of "drift networks" and "single score networks" lacks clear motivation and context. It is unclear why these are required, how they work, and what role they play in the overall approach. More motivation and context are necessary.
- **Choice of Hyperparameters**: The choice of $k=2$ (l. 216) as a window size lacks sufficient explanation. Similarly, the roles of bias and variance (l. 215) are not adequately explained.
- **Computational Cost**: The computational complexity of multi-marginal OT and spline calculations may limit the scalability of this approach for large datasets, which could be problematic for real-world applications---a more comprehensive discussion is required.
- **Competitors**: The paper considers a limited set of benchmark algorithms in its comparisons. Expanding the selection of competing methods would provide a clearer context for assessing the practical performance of MM-SF2M.
- **Experimental Setup**: The experimental setup lacks clarity, detailed motivation, and clear reasonings. It is unclear why specific datasets were chosen, what the intended outcomes of these experiments are, or how they systematically test the capabilities or limitations.
- **Datasets**: As experiments are limited to three synthetic datasets and only one real-world dataset, it is uncertain whether MM-SF2M can be broadly applied across diverse problem domains.
- **Pairwise vs. Triplet Model**: Figures 1–3 suggest that the pairwise model is quantitatively sufficient, which the authors also acknowledge. This behavior is quite unexpected, and additional investigations and clarifications are needed.
- **Mini-Batch Sampling**: While the authors mention inconsistency through mini-batch sampling, it has never been motivated why we would use mini-batch sampling in in the first place. A discussion is quite important, especially if they suspect a problem with this choice.
- **Experimental Conclusions**: Some experimental claims are unclear, for example the claim that the "GAE provides a consistent and invertible representation, allowing new data to be embedded in the same coordinate system **without distortion**." Given that distortion-free low-dimensional embeddings are highly challenging and quite unlikely, this point requires further proof, more substantiation, and considerable clarification.
- **Claims Not Validated**: The paper claims that the method enhances robustness and stability in multi-marginal settings "even when handling arbitrary timepoints", but lacks sufficient theoretical or experimental evidence to support this claim.

#### Minor Comments
- To enhance the appeal to a wider audience, the authors could consider broader terminology, such as using "relevant information" instead of "biological information."
- **Figures**: While pretty, Figures 1–3 are not particularly informative and could be compactified to make room for an extension to Section 2.
- l. 142: "challenging in high dimensions [CITATION NEEDED]": A reference here would support this claim, which, while likely true, would benefit from citation for readers interested in further exploration.

**Questions:**

-/-

---

> ### Author Response · Authors · 2024-11-25
>
> Thank you for your feedback. We have uploaded a rebuttal revision with deletions highlighted in gray and new additions highlighted in red.
>
> We address the weaknesses in our revision:
>
> 1. We have restructured our Introduction to better contextualize our work and describe our motivations and high level reasoning for our approach. Specifically, we consider Optimal Transport (OT) to be the Least Action Principle, which we discuss in Section 1.1. We further discuss transport splines in Section 2.1 and mention that we reformulate this as a composition of OT plans for i) mathematical convenience as our Loss function requires sampling from $z \sim \pi$ for some OT plan $\pi$, and ii) numerical implementation described in Section 2.3.2.
>
> 2. We include a more detailed discussion on the choice of window size $k$ in Section 2.3.1, which includes a discussion on the extra computational cost and scalability. We have also refined our notion of bias v. variance in terms of the conditional flows captured by a spline over a window. In particular, we describe in Section 2.3 that the rolling windows allow for variations in the spline segment connecting two endpoints, naturally generating a variance of conditional flows. This is in contrast to pairwise OT (which in effect considers a linear piecewise interpolation), which has less variation in the conditional flows. We include a Figure 2 in Appendix A where we highlight how the overlapping splines can generate multiple different flows interpolating two endpoints.
>
> 3. We include a brief discussion on cubic splines in Section 2.1 because the specific spline algorithm and window size $k$ are interrelated with respect to the conditional ODE flows. For clarity, we have instead moved the details of the natural cubic spline and monotonic Hermite cubic spline to Appendix A, along with a visualization of how spline algorithm affects the computed spline for varying timestep intervals.
>
> 4. We have updated our Results Section to better describe why we chose the datasets used and why we use mini-batch OT. We include results from two high dimensional gene expression settings in the CITEseq and Multiome datasets, validating our approach in terms of results and scalability. We also included further descriptions about the datasets and experimental setup in Appendix B.
>
> 5. We have clarified how we use "drift networks" and "single score networks" with respect to modeling the Fokker-Planck equation at the end of Section 2.3.3. We have also clarified the "same coordinate system without distortion" comment on GAEs by replacing the description with "shared coordinate system" in Section 3.3.

---

> > ### Comment · Reviewer_3yuc · 2024-11-26
> >
> > Thank you for taking the time to consider and incorporating some of my feedback.
> > After having reviewed your revision and response, I have revised my score accordingly.

---

> > > ### Author Response · Authors · 2024-11-28
> > > **Focused Follow-up Responses**
> > >
> > > We greatly appreciate your feedback in helping us clarify our manuscript.
> > >
> > > We include below some more pointed responses:
> > >
> > > Regarding Problem Formulation and Methodology:
> > > - We appreciate your observation about the need for clearer motivation of our methodological choices. We have fundamentally restructured our presentation to ground our approach in the Least Action Principle. The revised Section 1.1 now establishes how Optimal Transport naturally emerges as an implementation of least action, explains why pairwise transport between consecutive timepoints is suboptimal in multi-marginal settings, and demonstrates how measure-valued splines minimize total action $\int_0^1 \|\ddot{\mu}_t\|^2 dt$. This theoretical foundation provides clear motivation for each subsequent methodological choice, making the transitions between sections more natural and better justified.
> > >
> > > - Chen, Conforti, and Georgiou (2018) show that measure valued splines, defined as the measure valued curve minimizing the acceleration over processes $P$ given by $\inf_P \int_0^1 \int_\Omega \| \partial_{tt} X_t \|^2 dP dt$, admit optimal solutions $\hat{P}$ in Theorem 3.1. We are interested in these multi-marginal optimal transport splines from recognizing that these splines induce smoother transport compared to linear piecewise OT due its continuous differentiability with respect to time. Further, we hypothesize that this smoothness provides more stability and adaptability to irregular timepoints.
> > >
> > > Regarding Hermitian Cubic Splines:
> > > - Your concern about the apparent complexity introduced by cubic splines helped us realize the need for better organization of this material. While spline choice critically affects ODE flows, we agree the detailed derivations were distracting from the main narrative. We have moved the technical details to the supplemental materials Appendix A, while retaining essential intuition in Section 2.3.3. We've added Figure 2 specifically to demonstrate how different spline algorithms handle varying time intervals, making the practical importance of this choice clear. This reorganization helps maintain focus on key concepts while ensuring technical completeness.
> > >
> > > Regarding Drift Networks and Score Networks:
> > > - We thank you for highlighting the need for better motivation of these components. We have strengthened Section 2.3.2 to explicitly connect these networks to the Fokker-Planck equation. We show how they implement the reparameterization $u_t(x) = u_t^{\circ}(x) + \frac{g^2(t)}{2}\nabla \log p_t(x)$, and demonstrate why this separation enables efficient training. The revised presentation makes clear why both networks are necessary and how they work together.
> > >
> > > Regarding Hyperparameter Selection and Computational Cost:
> > >
> > > Computational complexity:
> > >   - For $M$ timepoints and window size $k$, we have $M-k$ possible windows. Thus, each training iteration costs $\mathcal{O}((M - k)c)$ where $c$ is the spline computation cost. We use monotonic Hermite cubic splines, which has a computational complexity of $\mathcal{O}(k)$ per spline. The complexity per training iteration becomes $\mathcal{O}((M-k)k)$ which is maximized at $k = M/2$ and minimized at $k=1$ and $k=M - 1$.
> > >
> > > Window Size Selection:
> > >   - We select a window size of $k=2$ so our per training iteration computational complexity is $\mathcal{O}(M)$. Moreover, setting $k=2$ involves splines over 3 control points, which is the minimum points needed for acceleration matching (at an intermediate control point) over two intervals. While $k > 2$ can potentially match a linear interpolation better than $k=2$, we opt for $k=2$ because the variation of flows provided by the overlapping windows offers better robustness of the learned flow. Moreover, additional intervals in the window do not meaningfully increase the information contained in monotonic Hermite cubic splines as these splines are very local, only depending on the immediately neighboring control points. We include a more in-depth discussion in Appendix A.
> > >   - We include a table of ablation studies examining the Pairwise, Triplet, and All ($k = M - 1$) models in Appendix C. These results empirically validate that increasing the window size $k$ does not result in any meaningful performance gain.
> > >
> > > Regarding Experimental Setup:
> > > We have substantially expanded our experimental validation:
> > >   - We have added high-dimensional tests on CITEseq and Multiome data
> > >   - Included detailed dataset descriptions in Appendix B
> > >   - Provided clear motivation for dataset selection
> > >   - We have added comprehensive ablation studies on the performance of our method for varying held-out time points on the S and $\alpha$-shaped datasets
> > >
> > > Regarding Mini-batch Sampling:
> > > We have added explicit justification for mini-batch OT:
> > >   - Computational efficiency considerations
> > >   - Theoretical guarantees on convergence to OT from Fatras et al. (2019, 2021)

---

> > > > ### Author Response · Authors · 2024-11-28
> > > > **Focused Follow-up Responses (cont.)**
> > > >
> > > > Regarding GAE Claims:
> > > > We appreciate your scrutiny of our GAE discussion. We have:
> > > >   - Replaced "without distortion" with more precise language
> > > >   - Clarified the role of shared coordinate systems
> > > >
> > > > These improvements are integrated throughout the manuscript, particularly in:
> > > >   - Revised theoretical foundations (Section 1.1)
> > > >   - Expanded methodology discussion (Section 2)
> > > >   - Enhanced experimental results (Section 3)
> > > >   - New appendices with detailed technical analysis
> > > >
> > > >
> > > > Thank you for your time in reviewing our work. We greatly appreciate the feedback in helping us clarify our manuscript.

---

### Official Review · Reviewer_uaar · 2024-11-03

**Soundness:** 3
**Presentation:** 3
**Contribution:** 2
**Rating:** 5
**Confidence:** 4

**Summary:**

This paper proposes a multi-marginal stochastic flow matching framework to align high-dimensional snapshot data observed at irregular time points. By extending simulation-free conditional flow matching methods, incorporating overlapping subsets, and using measure-valued splines, the authors aim to improve robustness and stability. The method is applied to synthetic datasets and single-cell biological data, showing potential for high-dimensional data alignment.

**Strengths:**

Strengths:
- **Originality**: This paper proposes a new method to improve multi-marginal stochastic distribution matching in a rolling window fashion.
- **Clarity**: This paper is well-written and easy to follow.
- **Significance**: The problem investigated in this paper is important in single-cell computational biology.

**Weaknesses:**

Despite the strengths discussed above, there still exists some concerns:

- **Methodological novelty** :  While this paper proposes a multi-marginal stochastic distribution matching framework by leveraging overlapping subsets and score matching, many of the novelty claimed in the paper are inherent over existing methods (e.g., SF2M [1], CFM [2, 3]). For example, coupling the distribution without reducing dimensionality (already be presented in both CFM and SF2M)， score matching (already be presented in SF2M), and measure-valued spline used in the computation of multi-marginal optimal transport. And handling the multi-marginal distribution matching can also be done by CFM and SF2M. The key novelty here, to my knowledge, is insert multi-marginal OT in SF2M framework with a rolling windows fashion, which may lack significant technical or methodological improvements.

- **Experimental design**:  There is a lack of direct comparisons with highly closely related methods like SF2M or CFM where this framework builds upon them, which undermines the credibility of the benchmarking.  Furthermore,  in SF2M it can handle thousands of gene dimensions, but this paper only tested 15 gene dimensions in the scRNA-seq data. It also raises the question of why the authors chose to compare their method with MIOFlow, which may rely on a low-dimensional manifold and seemed not be a fair enough comparison.

- **Weaknesses in explaining why the framework is better**: For example, the explanations of the claimed robustness and stability in multi-marginal settings and handling irregular time points are descriptive rather than supported by theoretical rigor. The paper claims to handle irregular time points effectively, but the main challenge in such scenarios is the information loss over longer intervals, which may not be easily mitigated by the algorithm alone. The paper does not demonstrate a concrete advantage in this regard.

- **Computational complexity and scalability**： When taking the multi-marginal OT in a rolling windows fashion rather than direct two point OT, there is a lack of discussions of the increased computational complexity and the balance between computational cost and accuracy. I think it may be taken into account to ensure a fair comparison.

- **Hyperparameter selection**: The hyperparameter $k$ seems an important factor in the framework. However, how to choose $k$ was not discussed in this paper, e.g., to balance accuracy and computational cost. It seems also need further investigations.

- **Code and data availability**: The code and data is not available.

Therefore I cannot recommend this paper for acceptance on this occasion but I would like to increase the score if the concerns are adequately addressed by the authors.

[1]  Tong, Alexander, et al. "Simulation-free schrödinger bridges via score and flow matching."  [2] Lipman, Yaron, et al. "Flow matching for generative modeling." [3] Tong, Alexander, et al. "Improving and generalizing flow-based generative models with minibatch optimal transport."

**Questions:**

see my questions asked in weaknesses.

---

> ### Author Response · Authors · 2024-11-25
>
> Thank you for your feedback. We have uploaded a rebuttal revision with deletions highlighted in gray and new additions highlighted in red.
>
> We address the weaknesses in our revision:
>
> 1. We have restructured our Introduction to better contextualize our work and highlight our motivations and high level reasoning for our approach. The sequence of Lipman et al. (2022), Tong et al (2023c) and Tong et al. (2023b), our work, and many to follow are is a line of exciting work that contributes to generative modeling, application in single cell data analysis, diffusion models, etc. Our effort was to delineate this sequence of contributions and also explain where our contributions come in and focus on developing a robust extension to irregular time multi-marginal data.
>
> 2. We have also restructured our Results section with additional results from high dimensional data for a few purposes:
> * Describe our data and why we chose this data. We follow up this discussion in more depth in Appendices B.2, B.3, and B.4.
> * We include a description of why we include MIOFlow in the Experimental Setup (Section 3.1).
> * We include a discussion on why our method is preferable, the window size $k$ selection, and how it scales to higher dimensionalities, We include an updated discussion on our method in Section 2.3 and a more detailed handling of the hyperparameter selection in Section 2.3.1. We note that although the method is agnostic to the spline algorithm, the specific choice does influence results, which we discuss in Appendix A. Scalability is also empirically validated with results from the higher dimensional CITEseq and Multiome datasets, shown in Table 2.

---

> > ### Comment · Reviewer_uaar · 2024-11-26
> >
> > Dear Authors,
> >
> > Thank you for your thoughtful response and the considerable time and effort you have invested in revising your manuscript. I appreciate the additional background information you have included, which enhances the context of your work.
> >
> > While I acknowledge the improvements made in the manuscript, my major concerns remain unresolved. I would like to outline these concerns in detail:
> >
> > - To my understanding, the primary novelty of your work lies in integrating multi-marginal Optimal Transport (OT) into the SF2M framework using a rolling windows approach. This combination seems straightforward. Could you clarify how this integration offers significant advancements over existing frameworks or what abilities have been achieved that previous algorithms could not?
> >
> > - It remains unclear why and how your algorithm effectively manages irregular time points. The manuscript lacks an intuitive explanation or conceptual understanding of this capability. From the experimental results, it seems that the algorithm performs better in terms of certain metrics; however, there is an absence of illustrative examples demonstrating scenarios where your framework significantly outperforms others. For instance, situations where other methods fail or produce incorrect results, whereas your algorithm succeeds, would provide compelling evidence of its effectiveness.
> >
> > - The only baseline presented in the manuscript is MIOflow.  It is puzzling why comparisons were not made with SF2M, CFM, OT-CFM, and other relevant frameworks, especially since your algorithm builds upon these foundations. Including such comparisons would offer a more comprehensive evaluation of your method’s performance and highlight its relative strengths and weaknesses.
> >
> > - The replies provided seem indirect and do not directly address the specific concerns raised. A more point-by-point, direct response to each of the issues highlighted would greatly enhance the clarity and effectiveness of your revisions.
> >
> > Thank you once again for your efforts.
> >
> > Best regards

---

> > > ### Author Response · Authors · 2024-11-28
> > >
> > > Thank you for your detailed feedback. We appreciate your thorough analysis that has helped us better articulate our contributions and strengthen our empirical validation.
> > >
> > > 1. Regarding Methodological Novelty:
> > > While the simulation free aspects of our method builds on SF2M and CFM frameworks, our contribution stems fundamentally from Schrödinger's original formulation and the Least Action Principle: the optimal path between multiple marginals should minimize the total action $\int_0^1 \|\ddot{\mu}_t\|^2 dt$ along the entire trajectory, not just between pairs (this is the extra action that deviated the transport from a mere Brownian motion). The consideration of global action minimization rather than pairwise transport helps our method naturally handle non-uniform time intervals.
> > >
> > > Chen, Conforti, and Georgiou (2018) show that measure valued splines, defined as the measure valued curve minimizing the acceleration over processes $P$ given by $\inf_P \int_0^1 \int_\Omega \| \partial_{tt} X_t \|^2 dP dt$, admit optimal solutions $\hat{P}$ in Theorem 3.1. We are interested in these multi-marginal optimal transport splines from recognizing that these splines induce smoother transport compared to linear piecewise OT due its continuous differentiability with respect to time. Further, we hypothesize that this smoothness provides more stability and adaptability to irregular timepoints.
> > >
> > > This foundation leads us to handle the key challenge of the probability path $p_t$ under dual constraints: it must both interpolate the distributions ("spatial" constraint) and respect specific timing of observations ("temporal" constraint).
> > >
> > > Previous approaches focused on pairwise transport, effectively minimizing $\sum_{i=1}^{M-1} W_2 ^2 (\mu_{t_i}, \mu_{t_{i+1}})$. Our innovation lies in:
> > >   - Using measure-valued splines to minimize total action $\int_0^1 \|\ddot{\mu}_t\|^2 dt$ across all timepoints
> > >   - Allowing different windows to model flow within intervals differently based on local dynamics
> > >   - Capturing variations impossible in prior models that only consider isolated intervals
> > >   - Smoothing flows at control points in Wasserstein space. This approach enhances robustness in both probability measure space and temporal evolution
> > >
> > > 2. Experimental Design and High-Dimensional Testing: We have substantially expanded our empirical validation, particularly testing robustness to challenging timepoint allocations. Key additions include:
> > >   - Results on highly irregular timepoint allocation $\mathcal{T}_3 = (0, 0.2, 0.27, 0.3, 0.88, 0.98, 1)$, which contains a short interval followed by a long interval into another short interval $(0.27, 0.3, 0.88, 0.98)$
> > >   - Demonstration of how our method adapts to changing velocities required by varying interval sizes
> > >   - Direct comparison with SF2M (implemented as $k=1$ case) showing superior performance particularly on S-shaped Gaussians where our method successfully learns to modulate speed
> > >   - Additional high-dimensional validation on CITEseq and Multiome datasets (up to 1000 dimensions)
> > >   - Comprehensive metrics ($W_1$, $W_2^2$, MMD(G), MMD(M)) across all timepoints
> > >
> > > 3-4-5.) Framework Robustness, Complexity, and Window Size Selection: The robustness of our framework emerges from both theoretical and practical considerations:
> > >
> > > Computational Analysis:
> > >   - For $M$ timepoints and window size $k$, we have $M-k$ possible windows. Each training iteration costs $\mathcal{O}((M - k)c)$ where $c$ is the spline computation cost. Our approach of using monotonic Hermite cubic splines gives $\mathcal{O}(k)$ per spline. Total complexity becomes $\mathcal{O}((M-k)k)$. This is worst when $k = M/2$ and optimal at $k=1$ or $k = M - 1$.
> > >
> > > Window Size Selection:
> > >   - We choose $k=2$ which still achieves a linear $\mathcal{O}(M)$ complexity. Moreover, $k=2$ provides the minimum number of points needed for acceleration matching at the interior control point of the spline.
> > >   - $k>2$ does not meaningfully increase information contained in splines because monotonic Hermite cubic splines are very local, only depending on the immediately neighboring control points. We include a discussion in Appendix A.
> > >
> > >   - We include a table of ablation studies empirically validating diminishing returns for $k>2$ in Appendix C.
> > >
> > > 6. Code and Data: The anonymous policy of ICLR does not let us share the GitHub repo at this stage. However, we are preparing to release our code and data in the camera ready version. The implementation follows the algorithm detailed in our paper, and we will provide comprehensive documentation and examples.
> > >
> > > These improvements appear throughout the revised manuscript, particularly in:
> > >   - Expanded theoretical foundation (Section 1.1)
> > >   - Detailed complexity analysis (Section 2.3.3)
> > >   - New results on high-dimensional datasets (Section 3)
> > >
> > > Thank you again for your comments, time, and effort in helping us clarify our manuscript.

---

> > > > ### Comment · Reviewer_uaar · 2024-11-28
> > > >
> > > > Thank you for your thoughtful and detailed response. I appreciate the additional information, which has given me a deeper understanding of the improvements you’ve made.
> > > >
> > > > - While I agree that considering a multi-marginal approach for global optimization is important, it appears that your work is incremental, to my understanding primarily using the multi-marginal OT  to replace the original OT within the SF2M framework. In the original SF2M paper, solving a Schrödinger bridge problem was theoretically justified. In your current work, there doesn’t seem to be theoretical proof demonstrating that introducing multi-marginal OT effectively solves a global action minimization problem. Could you clarify whether your approach indeed addresses such a problem, and if so, provide any theoretical guarantees?
> > > >
> > > > - From the numerical experiments presented, it seems that using a multi-marginal approach (k=2) does not yield significantly better results compared to the original SF2M (k=1). While there are some improvements in certain metrics, they do not appear to be substantial and stable. At this stage, it’s unclear to me in which specific scenarios multi-marginal SF2M offers clear advantages over the original SF2M, and why it can achieve that. Currently, it seems that a solid justification for these improvements has not been provided. Understanding the particular contexts or applications where your proposed method excels would greatly help in assessing its practical value.

---

### Official Review · Reviewer_tctL · 2024-11-04

**Soundness:** 3
**Presentation:** 2
**Contribution:** 3
**Rating:** 5
**Confidence:** 4

**Summary:**

The paper tackles the problem of modeling time evolution in high-dimensional systems, from limited snapshots at irregular time-points. In contrast to traditional dimensionality reduction approaches (which can oversimplify dynamics), this paper extends the recently proposed flow matching method to the multi-marginal setting. By learning flows over rolling windows of aligned subsets of snapshots, they focus on modeling local dynamics with additional improvements to incorporate stochasticity and reduce overfitting.

The method is validated on three synthetic datasets, constructed with specific characteristics, and a real world dataset of cancer cell lines from uneven timepoints.

**Strengths:**

**Significance**
The problem being studied is highly significant, given its utility in understanding biological processes, and designing effective interventions

**Originality**

Since the paper focuses on extending flow matching to the multi-marginal setting, my comments are restricted to this. Currently, flow matching methods focus on learning vector fields that transport a distribution $q_0$ towards a distribution $q_1$, without explicit modeling of $q_t$. Extending flow matching to multi-marginal settings can be accomplished by either using trajectory information (if available), or performing flow matching over restricted subsets of snapshots. This paper adopts the latter route.

The paper locally aligns snapshots, and uses this alignment to compute the conditioning vector fields for flow matching. The proposed solution is simple and elegant in that regard, and easy to implement computationally.

**Clarity**
The paper in itself is well-written and clear, and the motivation, objectives and method were easy to follow. Some of the presentation can be enhanced (see Weaknesses and Questions), especially in experiments.

**Quality**
The quality of the submission is moderate. The proposed idea is simple and well-executed, the text is well-written, but presentation can be improved in the experiments and prior work needs to be properly contextualized.

**Weaknesses:**

The main weaknesses of the papers are in the presentation concerning the experimental section, and improper contextualization of prior work.

**Prior Work**

OT-based methods for understanding cellular modelling of trajectories are largely missing. Example such works included below:
1. Optimal-transport analysis of single-cell gene expression identifies developmental trajectories in reprogramming. Schiebinger et al 2019.
2. Proximal Optimal Transport Modeling of Population Dynamics. Bunne et al 2022.
3. Trajectorynet: A dynamic optimal transport network for modeling cellular dynamics. Tong et al. 2020

Multi-marginal generative modeling has also been studied by a few works in recent times:
1. Deep multi-marginal momentum schrödinger bridge. Chen et al 2023
2. Multimarginal generative modeling with stochastic interpolants. Albergo et al. 2023

I can see the methodological differences between existing work and the current one, but without proper experimental comparison, it makes it unclear why and where the proposed method improves / fails compared to the previous ones.

**Experiments**

The experiments of synthetic data would benefit from improved presentation. My takeaway is that the authors validation attempts to recover bifurcation points in the trajectories, based on other local snapshots but this is very hard to see in the figures.

1. It would be nice to add a colored window / vertical line to differentiate between training and validation snapshots, and improve the font corresponding to time.
2. The authors should also add the ground truth trajectories wherever possible, or make the ground truth snapshots more visible.

**Questions:**

**Experiments**
1. On the DynGen simulated trajectories, MIOFlow seems to do a much better job at capturing the evolution compared to the proposed method. Do the authors have any insight on whether this happens because of a specific training-validation split or an inherent difference between the two methods?

2. The experimental validation can further be improved by considering multiple snapshots to predict instead of just a single one. For example, in the $\alpha$-gaussians, it would be interesting to see if the model can recover the bifurcations given just the snapshots in the initial and latter timepoints.

3. Is local modeling of dynamics always the best option? I agree there must be a tradeoff between local vs global dynamics modeling, but my intuition is that encouraging the model to learn global patterns (by doing flow matching on initial and terminal distributions), while having rolling windows locally would help the model learn both local and global patterns. I would be interested in hearing any comments the authors have in this regard, and if they have some experimental validation for this.

---

> ### Author Response · Authors · 2024-11-25
>
> Thank you for your feedback. We have uploaded a rebuttal revision with deletions highlighted in gray and new additions highlighted in red.
>
> We address the weaknesses and questions in the following ways:
>
> 1.  We have restructured our introduction to include a more comprehensive literature review further highlighting our motivations and high level reasoning for our approach. We are continuing a sequence of exciting works Lipman et al. (2022), Tong et al. (2023c), Tong et al. (2023b) for generative modeling and trajectory inference for use in domains such as single cell data.
>
> 2. The performance difference on the DynGen dataset for MIOFlow and our method can be traced to how the DynGen dataset was preprocessed via PHATE into 5 latent dimensions. We include a discussion of how this benefits MIOFlow in Appendix B.2.
>
> 3. To address local v. global dynamics we have included a general discussion on our choice of the window size $k = 2$ in Section 2.3.1 to address concerns about the influence of this hyperparameter in terms of the conditional flows and the computational costs. We have also added a further discussion on how the choice of spline algorithm affects the spline under varying time intervals for $k=2$ in Appendix A. Our hybrid approach of using overlapping windows allows us to leverage local windows over varying timescales constrained by global boundary conditions.

---

> > ### Comment · Reviewer_tctL · 2024-11-26
> > **Thank you for your Response**
> >
> > I thank the authors for the time and effort taken in the response.
> >
> > I generally like the idea proposed in the paper, and it has merit. That said, the paper has undergone extensive rewriting, and is now over the 10 pages limit (15 actually), and in my opinion is not currently ready for acceptance at ICLR. Its also unclear if the visualizations have changed (as mentioned in the Weaknesses section), and the visualizations are still hard to follow.
> >
> > I will keep my score as is.

---

> > > ### Author Response · Authors · 2024-11-28
> > >
> > > Thank you for your insightful feedback, particularly regarding prior work contextualization and experimental validation. We have made substantial improvements to the manuscript based on your comments.
> > >
> > > We follow up with a more in-depth response below:
> > >
> > > 1) Regarding Prior Work: We have expanded our literature review to better position our work within the broader context of OT-based methods for cellular modeling. As you noted, important works like Schiebinger et al. (2019) and TrajectoryNet (Tong et al., 2020) pioneered OT applications in this domain. However, these approaches typically operate in reduced dimensions as they rely on simulating trajectories through numerical integration.
> > >
> > > Our key distinction lies in how we handle the multi-marginal setting:
> > >
> > > a) Previous methods fall into three categories:
> > >   - Traditional OT methods using simulation-based approaches requiring dimensionality reduction (Trajectorynet, Bunne et al. 2022)
> > >   - Recent deep learning approaches like Chen et al. (2024) and Albergo et al. (2023) that work in high dimensions but require expensive flow integration and memory-intensive trajectory caching during training
> > >   - Flow matching methods that consider only pairwise transport between consecutive timepoints (Lipman et al., 2022; Tong et al., 2023)
> > >
> > > b) Our approach:
> > >   - Works directly in the ambient high-dimensional space through simulation-free flow matching
> > >   - Follows Schrödinger's original principle by minimizing total action $\int_0^1 |\ddot{\mu}_t|^2 dt$ across all timepoints, which by Occam's razor yields the simplest flow connecting the marginals
> > >   - Uses overlapping triplets ($k=2$) to optimally balance computational efficiency ($\mathcal{O}(M)$ complexity) with acceleration matching needed for proper spline construction
> > >   - Avoids expensive trajectory generation or caching during training while maintaining high-dimensional structure
> > >
> > > 2. Regarding Experimental Presentation: We have thoroughly revised our experimental section to:
> > >   - Improve overall coloring to differentiate between different timepoints
> > >   - Add borders to differentiate between experiments
> > >   - Improve time labeling and ground truth visualization
> > >   - We include validation across multiple held-out timepoints in Appendix C.
> > >   - We discuss the bifurcation performance along with the DynGen dataset below.
> > >
> > > 3. Regarding DynGen Performance:
> > > While the DynGen visualization in reduced dimensions appears to favor MIOFlow, we emphasize that analyzing dynamics in the original high-dimensional space is crucial for capturing biological heterogeneity. Low-dimensional projections can artificially collapse distinct cell states and trajectories, obscuring important biological variation. Moreover, the DynGen dataset which we adopt from (Huguet et al. 2022) preprocesses the data using PHATE (Moon et al. 2019) which encodes the data onto a manifold with a Gaussian kernel metric. This aligns with MIOFlow's assumption of learning trajectories on a diffusion-based manifold. We include this discussion in Appendix B.2.
> > >
> > > Our method maintains and analyzes the full high-dimensional structure, as demonstrated by superior performance on raw CITEseq and Multiome data where preserving cellular heterogeneity is essential.
> > >
> > > 4. Regarding Local vs Global Dynamics:
> > > Your question about balancing local and global patterns touches on a fundamental aspect of our approach. Our rolling window strategy with $k=2$ achieves this balance by:
> > > - Capturing local dynamics through overlapping triplets
> > > - Maintaining global consistency via shared timepoints between windows
> > > - Matching acceleration between consecutive intervals through second-order spline dynamics
> > > - We visualize how the choice of spline algorithm influences local dynamics, and how the overlapping windows can capture flow variations while maintaining global consistency in Figure 2.
> > >
> > > The measure-valued spline approach naturally enforces smoothness constraints while allowing local variation by minimizing the integrated squared acceleration $\int_0^1 \|\ddot{\mu}_t\|^2 dt$.
> > >
> > > These improvements appear throughout the manuscript, particularly in:
> > >   - Expanded literature review (Section 1.2)
> > >   - Enhanced experimental results (Section 3)
> > >   - New appendix sections detailing dataset preprocessing and visualization
> > >   - Additional comparisons on higher dimensional datasets and more difficult timepoints $\mathcal{T}_3$. The changes maintain our core contribution of efficient high-dimensional multi-marginal transport while providing clearer empirical validation and theoretical context.
> > >   - Enhanced visualizations (Figure 1 and Appendix D)
> > >
> > > We appreciate your time and again thank you for pointed feedback in clarifying and sharpening the manuscript.

---

### Official Review · Reviewer_Zq1o · 2024-11-11

**Soundness:** 4
**Presentation:** 2
**Contribution:** 2
**Rating:** 6
**Confidence:** 2

**Summary:**

The paper presents a novel flow matching method. This method can be applied to high dimensional data measured at non-equidistant time points without reducing dimensionality, so that the dynamics of the data are not oversimplified. The method is validated on several datasets and showed the improvement compared to existing methods.

**Strengths:**

I think the paper is in good quality. Other than the concerns about the window size k, the paper is clearly written.

**Weaknesses:**

My impression is that this paper is somewhat an extention of Tong et al. (2023b). Although, my research is not in stochastic flow matching, so I cannot correctly evaluate originality or significance.

As far as I understand, the window size k is a hyperparameter that critically affects the performance. Though it is not clear how to choose k. On page 5, it seems like k=2 is a natural choice, but without any reasoning or discussion. I think a discussion of the tuning of k would be helpful to readers, for e.g., it should be just trying several different values of k and choose the best one, or k=2 is the natural choice for some reason.

On Table 1, triplet(k=2) seems to dominate pairwise(k=1) for many cases, which makes readers to wonder if k=3 can be even better. This is due to that "k is balancing bias and variance" in line 215, since then bias-variance tradeoff is likely to appear, i.e., as k increases, the cost would decrease and increase. If there is a reason to use k=1 or 2, a reasoning should be helpful, and if there is no specific reason to stick to k=1 or 2, then more experiments with increasing the values of k and observing the bias-variance tradeoff trend would be very helpful to readers.

Algorithm 1 is written for general values of k, but in line 216-276 the k is fixed as 2 in the explanation. My impression is that explaining with k=2 doesn't really simplify the explanation by much, but making extra confusions: I wondered if the explanation would be the same for general k or some extra changes are needed. I think it's better to write the explanation in line 216-276 with general k.

--------------------
Thanks to the authors for reflecting on my reviews. I still think that the weakness (in particular about the novelty) presents, but now I can understand the authors' contribution. Hence I increase the rating score from 5 to 6.

**Questions:**

As far as I understand, in line 9 and 10 of Algorithm, X_i should be X_{t_i}, based on the explanations on Section 2 3.1.. Are these typo, or is my understanding wrong?

---

> ### Author Response · Authors · 2024-11-25
>
> Thank you for your feedback. We have uploaded a rebuttal revision with deletions highlighted in gray and new additions highlighted in red.
>
> We address the weaknesses in the revision in the following ways:
> 1. Yes, this paper is an extension of Tong et al. (2023b). The sequence of Lipman et al. (2022), Tong et al (2023c) and Tong et al. (2023b), our work, and many to follow are is a line of exciting work that contributes to generative modeling, application in single cell data analysis, diffusion models, etc. Our effort was to delineate this sequence of contributions and also explain where our contributions come in and focus on developing a robust extension to irregular time multi-marginal data. To highlight this, we have restructured our introduction to provide a more comprehensive literature review and high level reasoning for our approach. Specifically, the **Our Approach** section details this reasoning.
>
> 2. We have included a general discussion on our choice of the window size $k = 2$ in Section 2.3.1 to address concerns about the influence of this hyperparameter in terms of the conditional flows and the computational costs. We have also added a further discussion on how the choice of spline algorithm affects the spline under varying time intervals for $k=2$ in Appendix A.
>
> The algorithm did indeed contain typos. We have updated the $X_i$s to be $X_{t_i}$s to be consistent with the prior descriptions we present.

---

> > ### Author Response · Authors · 2024-11-28
> >
> > Thank you for your thoughtful feedback. Your comments have helped us significantly improve the manuscript's clarity and technical rigor. We would like to follow up in more detail based on a final draft of our manuscript.
> >
> > 1. Regarding originality and relationship to Tong et al. (2023b):
> > Our work differs fundamentally from prior approaches in how we handle multi-marginal transport. While previous methods like Lipman et al, and Tong et al. (2023b) consider pairwise transport between consecutive timepoints, we recognize this does not yield optimal paths when multiple marginals are involved. Our key insight stems from Schrödinger's original formulation and the Least Action Principle: the optimal path between multiple marginals should minimize the total action $\int_0^1 \|\ddot{\mu}_t\|^2 dt$ along the entire trajectory, not just between pairs. Note that the consideration of the global action minimization rather than pairwise transport help our method naturally handle non-uniform time intervals.
> >
> > Chen, Conforti, and Georgiou (2018) show that measure valued splines, defined as the measure valued curve minimizing the acceleration over processes $P$ given by $\inf_P \int_0^1 \int_\Omega \| \partial_{tt} X_t \|^2 dP dt$, admit optimal solutions $\hat{P}$ in Theorem 3.1. We are interested in these multi-marginal optimal transport splines from recognizing that these splines induce smoother transport compared to linear piecewise OT due its continuous differentiability with respect to time. Further, we hypothesize that this smoothness provides more stability and adaptability to irregular timepoints.
> >
> > 2. Regarding window size $k$ and its selection:
> > The window size $k=2$ represents an optimal choice for several reasons:
> > * Acceleration-based justification: To construct a measure-valued spline minimizing $\int_0^1 \|\ddot{\mu}_t\|^2 dt$, we need at least 3 points ($k=2$ window) to properly capture the acceleration $\ddot{\mu}_t$ between consecutive timepoints.
> > * Computational efficiency: For $M$ timepoints and window size $k$, we have $M-k$ possible windows, each requiring $k$-point optimal transport computation. This gives complexity $\mathcal{O}((M-k)k)$. With $k=2$, we achieve linear $\mathcal{O}(M)$ complexity while capturing essential second-order dynamics.
> > * Empirical validation: We tested different window sizes and found increasing $k$ beyond 2 showed negligible returns while increasing computational cost. This aligns with the theoretical understanding that acceleration terms captured by $k=2$ suffice for characterizing key dynamics.
> >
> > 3. Regarding Algorithm explanation and k-consistency:
> > You correctly note that lines 216-276 focused on $k=2$ while Algorithm 1 was general. We have revised the manuscript in Section 2.3 to:
> > - Present the general algorithm for any $k$ first
> > - Provide specific implementation details for $k=2$ as our recommended configuration
> > - Add clear notation about time indexing ($X_{t_i}$ vs $X_i$)
> >
> >
> > The essential feedback helped us realize we needed to better motivate our key design choices, particularly regarding window size selection and algorithm presentation. These improvements appear in:
> > - New section on window size selection (Section 2.3.3)
> > - Expanded theoretical motivation (Section 2.1)
> > - Updated algorithm notation
> > - Added computational complexity analysis

---

### Note · Authors · 2025-06-29

I have read and agree with the venue's withdrawal policy on behalf of myself and my co-authors.

---

### Meta-Review · Area_Chair_E9zD · 2024-12-23

**Metareview:**

This paper uses stochastic flow matching ideas to align snapshots of trajectory data in discretely observed, possibly irregular settings. The paper has some technical elements, and the reviewers found many merits in this work, but also had remaining concerns that ultimately should be addressed with further review. Some also mention possible overlap with recent related work, which has partially been addressed in the rebuttal phase. The article contains promising ideas and will benefit from revision taking into consideration the constructive comments of reviewers here

**Additional Comments On Reviewer Discussion:**

reviews were helpful though limited discussion

---

### Decision · Program_Chairs · 2025-01-22

Reject